# Qualitative Investigation of Experience and Quality of Life in Patients Treated with Calcium Electroporation for Cutaneous Metastases

**DOI:** 10.3390/cancers15030599

**Published:** 2023-01-18

**Authors:** Kitt Vestergaard, Mille Vissing, Julie Gehl, Christina Louise Lindhardt

**Affiliations:** 1Centre for Nursing, Absalon University College, 4000 Roskilde, Denmark; 2Center for Experimental Drug and Gene Electrotransfer (C*EDGE), Department of Clinical Oncology and Palliative Care, Zealand University Hospital, 4000 Roskilde, Denmark; 3Department of Clinical Medicine, Faculty of Health and Medical Sciences, University of Copenhagen, 2200 Copenhagen, Denmark; 4Centre for Research in Patient Communication, Clinical Institute, Department of Health Sciences, University of Southern Denmark, 5000 Odense, Denmark; 5Academy of Geriatric Cancer Research (Age Care), Odense University Hospital, 5000 Odense, Denmark

**Keywords:** electroporation, cancer, qualitative interviews, calcium electroporation, quality of life, patient experience, electroporation based treatments

## Abstract

**Simple Summary:**

Some cancer patients develop cutaneous metastases at late stage of disease, and the tumours may be present for months or years. Some cancers are more prone to disseminating to the skin, such as breast and lung cancer. When cancer manifests on the body’s surface or on the face, symptoms and distress can have a significant impact on the quality of life of patients. Surgical and medical treatment is difficult, and local treatments are important. Calcium electroporation is a novel cancer treatment. It includes injecting calcium-solution and applying electric pulses to tumour tissue. There is a scarcity of literature on patient experience of living with cutaneous metastases. Our study’s objective was to examine the patient’s perspective on CaEP treatment of skin tumours, as well as the treatment’s impact on health-related quality of life (QoL). We evaluated quality of life in patients with skin metastases treated with calcium electroporation using qualitative interviews. Calcium electroporation enhanced health-related quality of life by reducing symptoms and increasing social inclination. Peer accounts are important in preparation for treatment.

**Abstract:**

(1) Background: Calcium electroporation is a novel cancer treatment. It includes injecting calcium-solution and applying electric pulses to tumour tissue. Data on quality of life for patients with cutaneous metastases treated with calcium electroporation is limited. We evaluated quality of life in patients with skin metastases treated with calcium electroporation using qualitative interviews. (2) Methods: This investigation featured a subgroup from a non-randomised phase II study (CaEP-R) at Zealand University Hospital, Denmark, studying response to calcium electroporation in cutaneous metastasis (ClinicalTrials no. NCT04225767). Participants were interviewed at baseline before calcium electroporation treatment and after two months. Data was analysed phenomenologically; (3) Results: Interviews were conducted February 2020–November 2021. Nine patients were included, of which seven participated in both interviews. All seven patients expected treated tumours to disappear, symptom relief and minimal side effects. Most patients requested peer accounts. All patients found the treatment uncomfortable but acceptable; all thought their fears of electric pulses exceeded their experience. All would repeat the treatment if effective. Successful treatment had a positive effect on pain, symptomatic wounds, sleep, vigour and social inclination; (4) Conclusions: Calcium electroporation enhanced health-related quality of life by reducing symptoms and increasing social inclination. Peer accounts provide patients with a shortcut to confidence in treatment on top of doctors’ recommendations.

## 1. Introduction

Although skin cancer is the most common cause of cutaneous malignancy, nearly all types of cancer can potentially metastasise to the skin [1]. Cutaneous metastasis is typically a symptom of advanced disease but can be present for months or even years [2]. For example, cutaneous metastases occur in 5 to 20% of breast cancer patients and are the most common type of cutaneous metastasis in women [2,3]. The most common cause of cutaneous metastasis in men is lung cancer, which has a 3% incidence of cutaneous metastasis. [2,3,4]. When cancer is present on the body’s surface or on the face, symptoms and distress can have a significant impact on the quality of life of patients [2,5]. Surgical and medical treatment is difficult, and local treatments are important [6].

There is a scarcity of literature on cutaneous metastases, particularly on patient experience of living with cutaneous metastases. This may indicate that the influence on quality of life, caused by cancer that has spread to the skin, could be underestimated.

Our study’s objective was to examine the patient’s perspective on CaEP treatment of skin tumours in patients with a variety of primary cancer diagnoses, as well as the treatment’s impact on health-related quality of life (QoL) [1].

Calcium electroporation (CaEP) is an effective, novel treatment for cancer in the skin [7,8]. Depending on the severity of the condition and patient request, the procedure may be conducted under local or general anaesthesia [9]. Calcium is supplied via intratumoural injection with a peripheral margin, followed by the direct application of electrical pulses to the target area. Electroporation generates transitory membrane holes that allow calcium to diffuse into otherwise impermeable target cells [10]. Due to high calcium levels, the targeted cancer cells may perish while normal cells re-establish homeostasis [11]. Calcium electroporation is an innovative anti-cancer treatment that often requires only one or a few treatments, requiring fewer patient visits than, e.g., radiation therapy. The treatment is safe, with low treatment toxicity and costs, and spares the patient from exposure to mutagens and damage of healthy tissues.

A non-randomised phase II multi-centre study, CaEP-R (ClinicalTrials.gov Identifier: NCT04225767) [1], has investigated response to calcium electroporation in patients with cutaneous or subcutaneous malignancy, with an enrolment of 19 patients across three centres. The study is a collaboration between three cancer centres in Denmark and Germany. The primary endpoint is tumour response to treatment after two months. Secondary endpoints include visualising response to treatment in a subset of patients using MRI and histological analyses of biopsies (optional) from treated areas performed after one year. Detailed materials and methods can be found in the published protocol (doi: 10.1136/bmjopen-2020-046779).

Qualitative interviews are an effective method for gathering comprehensive information and can be employed to explore almost any subject in depth [12]. Participants can contribute information in their own words and from their own perspectives as opposed to limited response options, as is the case in survey research [13]. Researchers can gather other useful information from the interview setting such as observation of body language or patient demands [14]. Since qualitative interviews yield extensive data, they are important for researching complexities of patient expectations, experience and quality of life [13,15]. This study uses a phenomenological approach, describing and explaining an event or phenomena, in this case cutaneous metastases and calcium electroporation treatment, from the perspective of patients who have experienced it [13].

## 2. Materials and Methods

The methods and materials used are thoroughly described in a protocol article, ClinicalTrials.gov Identifier: NCT04225767 [1]. Briefly, qualitative interviews were performed in a subset of patients treated in the CaEP-R study at Zealand University Hospital, Denmark. All patients received calcium electroporation treatment, and calcium electroporation was not compared to other treatment modalities. Patient could continue any concomitant systemic treatment. Interviews before—as well as two and twelve months after treatment—were conducted from February 2020 to November 2021. The first interview took place after the preliminary examination, before CaEP, the day of treatment.

This study took place during the COVID-19 pandemic. For the first two months, interviews were conducted at The Danish Cancer Society, close to the study site. The interviewer did not wear a uniform and did not participate in therapy. The interviewer was independent of the treatment facility. Two months into the project, interviews were conducted by telephone to reduce the risk of coronavirus contamination. This was maintained throughout the remainder of the study.

### 2.1. Participants

Patients were included in the CaEP-R study based on the following criteria: age older than 18 years; ability to understand the participant information; histologically verified cutaneous or subcutaneous cancer of any histology; previously offered other relevant standard treatment for their cancer disease; progressive or stable disease present after a medical treatment period of two months or more; ongoing radiation therapy not involving the CaEP treatment area; performance status ECOG/WHO ≤ 2; at least one cutaneous or subcutaneous tumour measuring up to 3 cm; if sexually active, use of safe contraception; signed informed consent [2]. Exclusion criteria were pregnancy or lactation [1].

### 2.2. Interviews

The same experienced qualitative researcher carried out all interviews. We used an external interview specialist and interviewed outside the hospital setting. With these precautions, we aimed to prevent bias from patients withholding facts due to embarrassment or concern for their therapy.

The interview guide was inspired by Kvale and Brinkmann [16]. The qualitative interview was semi-structured with open-ended questions. The guide allowed change during the interview to ensure a focus on unexplored topics. The questions examined patients’ emotional and physical well-being with emphasis on symptoms related to their cutaneous metastases, before and after CaEP treatment. Both face-to-face and phone interviews used interview guides. The recorded interviews were transcribed, processed and categorised using a phenomenological approach [17].

### 2.3. Data Analysis

The interviews gave descriptive data regarding patients’ views on CaEP treatment. A thematic data analysis was used to consider patients’ perspectives. This study uses a qualitative descriptive approach as a thematic analysis to examine a simple phenomenon [10]. Generated data was used for analysis and presentation [18]. Experiences were communicated in common language [17]. Analysis themes depended on researchers’ perceptions and preferences [12].

Braun and Clarke’s thematic analysis was applied in the following six steps [17]: first, two scholars read and reread the content to check for accuracy and typos; second, researchers debated the transcribed content, and similarities or variances were noted; third, data was coded; fourth, initial codes were examined and disclosed. The fifth step was thematic mapping, with adjustment of themes and sub-themes. In the sixth and last step, each concept was elaborated using existing literature and delineated. Data consistency ensured code placement in a theme. Each code’s quotes were reviewed for patterns. A co-author confirmed the study’s consistency and validity. Continual coding and context reviews provided consistency. A logbook was kept during the process [17].

### 2.4. Ethical Considerations

All participants gave informed consent. Personal data was stored according to good clinical practice and confidentially according to Danish recommendations. The CaEP-R trial is approved by the European Medicines Agency and Danish Medicines Agency, the Danish Regional Committee on Health Research Ethics, (SJ-810) and Data Protection Agency (REG-115-2019) and registered on EudraCT (2019-004314-34) and ClinicalTrials.gov (NCT04225767). A good clinical practice (GCP) representative monitored the study.

## 3. Results

### 3.1. Participants

Patients with a variety of cancer diagnoses participated. Nine patients were interviewed before treatment with CaEP. Interviews were conducted with seven of these patients two months after treatment (Table 1).

Seven of these patients were re-interviewed two months after treatment (see Table 1). The remaining two patients did not participate due to the progression of their primary cancer disease, which was unrelated to CaEP. Three patients were interviewed one year after treatment, and four patients had passed away. Given the small and biased cohort of cancer survivors, the qualitative analyses and resulting data were restricted to interviews conducted before and two months after CaEP treatment.

### 3.2. Themes and Subthemes

The overall theme of the interview guide was “calcium electroporation for patients with cutaneous metastases”. Questions at baseline mainly focused on expectations. Responses showed subthemes of previous experiences, need for peer accounts, hope for successful treatment outcome and concerns regarding electricity and pain (Figure 1). Questions in the interview two months after treatment elaborated on treatment and post-treatment experiences. Responses showed subthemes related to treatment experiences, side effects, symptom relief and the impact on quality of life. The subthemes are expanded upon with quotes below.

### 3.3. Cutaneous Metastases

The included tumours were dermal in location (arms, chest, groin, neck, face and scalp). Three patients reported that their tumours were ulcerated or bleeding. Fear of wound seeping or that visible tumours would alert others to their illness prevented three individuals from engaging in social activities outside the home. Four individuals experienced sleep disturbances due to localised pain. The following three quotes were expressed related to patients’ experience of cutaneous metastases:

“I wear a bra if I have to do something nice and the tumour is right where the bra would be. So I can’t wear a bra, and that’s why I don’t really go out.”Female, 68

“I have a (tumour) on my shoulder blade here at the back, so it’s a bit difficult to lie down properly without it hurting. It also feels very strange—it’s like a foreign body—It just has to go.”Male, 62

“They are shocking; they just come and come (…). They are not nice to look at. They bleed and ooze—some are like blisters and I use large special bandages that only last a day (…). I can’t bear to be visited by a nurse at home every day. I think it disturbs my husband as well. Even though we’ve known each other for a hundred years, it’s still like, damn, this isn’t fun.”Female, 62

### 3.4. Experiences with Previous Cancer Treatments

Before CaEP, all seven participants had extensive healthcare and cancer treatment experience. Patients notably voiced negative experiences from previous treatment courses and feelings of being unheard. All seven patients had previously experienced doctors not taking their personal situations from the medical record into account in their consultation and treatment plan or had been lacking information. These experiences taught the patients to be vigilant and proactive. In this setting, patients had pre-emptively worried that unsuccessful management or “human error” might prolong their treatment course. All seven patients had previously undergone chemotherapy, radiation therapy and/or immunotherapy. These medications’ adverse effects, such as nausea, vomiting and weakness, disrupted the patients’ daily lives. All had experienced feeling unwell during lengthy regimes and found it difficult to schedule and attend multiple treatment sessions. Patients were informed that CaEP was expected to be safe with low toxicity. Patients compared this information to their prior treatment experiences and their experiences with the healthcare system. All seven patients eagerly anticipated a treatment with minimal adverse effects.

“If this can become something that becomes more widespread for certain types of cancer, well… The course is shorter, you have to go to the hospital less times and interrupt your daily life less times (…) and you don’t have a lot of side effects or nausea, or like, hair loss. After all, there are also many different and delayed effects of radiation. That’s why I’m participating and said I’ll try it. Now I’m just crossing my fingers that they can keep it down. Compared to entering a course of chemotherapy again, well, I think it’s pretty great.”Female, 63

### 3.5. Expectations

Generally, patients were optimistic about the efficacy of the calcium-based therapy for their tumours. They all found it surprising that something as simple as calcium could be effective and expected the procedure to have few side effects compared to their past cancer treatments. All patients expected the treatment to be performed once or twice at most and to be gentler than other cancer treatments. Hope that the CaEP therapy would be effective included an expectation of anxiety and symptom relief from issues such as sleep disturbances due to localised pain, ulceration and bleeding, wound seepage and the ability to resume social activities without worry about visible tumours and stigma. One patient remarked,

“The treatment tomorrow is just a piece of cake. If it’s successful, the treated tumors will be gone. Maybe some others will show up, but they can be dealt with later. Removing the tumors will bring more peace of mind and make the immediate threat disappear.”Female, 71

Generally, patients were optimistic about treatment efficacy for their tumours. All the patients emphasised that they found it surprising that something as simple as calcium was an effective and central element of the treatment. All expected the procedure to have few side effects compared to their past cancer treatments. All patients expected the treatment to be performed once or twice at most and to be gentler than other cancer treatments.

Hope that the CaEP therapy would be effective included an expectation of symptom and anxiety relief from the following: sleep disturbances due to localised pain, ulceration and bleeding, wound seeping and refraining from social activities and worries about visible tumours and stigma.

“The treatment tomorrow, it’s just a piece of cake. Then the tumours are gone, well, those that are treated, they are gone. Then, maybe some others will show up, and they will be dealt with. And that’s it. Having the tumours removed will mean more peace of mind. The immediate threat disappears.”Female, 71

This was expressed by another participant

“I hope that the calcium electroporation treatment will get rid of the tumours so I can cycle and swim again”Female, 62

At the preliminary examination, the treating physicians informed the patients about how the treatment would take place. Each patient seemed to follow a similar pattern of asking about other patients’ treatment experiences.

“I wonder how bad it is to go through, and how much pain it will cause. Will it need lots of injections and such.”Female, 67

And addressed by another participant:

“The issue of the treatment are the electric pulses. They cause some muscle contractions. Of course, I understand that part rationally. However, the uncertainty about how the treatment feels in the body is scary. The doctors cannot explain this part, as they have not tried it. I cannot imagine the feeling of getting electric current through the body. Accounts on how the pulses during calcium electroporation feel from other patients would ease my mind. Especially if they say that it does not hurt.”Female, 63

### 3.6. Concerns

All seven patients were afraid that the treatment might involve electricity and cause discomfort prior to the procedure. Three patients talked about death and were emotionally upset in several instances.

While being a necessity for daily existence, people are aware that electricity can be hazardous or even fatal. Electrical power may be related with pain, cognitive changes, heart rhythm changes, danger or even death in other treatment scenarios. Concerns about electrical therapy is acknowledged in electroconvulsive therapy used in psychiatry [19]. Studies on laser therapy have also shown that patients fear electrophysical components of treatment [20].

“I am afraid to be awake while the treatment is taking place. I am not too fond of getting electric shocks, even small ones.”Female, 67

### 3.7. Treatment Experience

Even slight unpredictability can affect cancer patients [21,22,23]. The interior design, scent and location of the facility affected the patient’s holistic treatment experience. Other studies have stated the importance of treatment atmosphere [24,25]. During the study, adjustments were made to patient reception and facilities.

All patients reported that their concerns regarding electric pulse-therapy far exceeded the reality. No patients reported any experiences from calcium injections and were unaffected. A few patients noted that there were many needles involved in the treatment (e.g., needles used for local anaesthesia and calcium injection as well as the needle electrode). Three patients expressed their experience as follows:

“The electric pulses in the body do not feel pleasant, but they are bearable.”Female, 67

or,

“Despite the discomfort related to the treatment, I would have done it again.”Female, 62

and,

“I believe the nervousness would have been less if I had heard from other patients that the calcium electroporation treatment is tolerable for pain.”Female, 68

### 3.8. Side Effects

Overall, patients reported few side effects. The most frequently occurring side effects were transient soreness (experienced by 5/7 patients, lasting up to 14 days) and ulcerated wounds (experienced by 5/7 patients). Two patients experienced transient wound odor for a few days, and one patient each reported numbness and transient odor for a few days. All patients reported slight darkening or redness of the skin two months after the CaEP treatment (see Figure 2). One patient said,

“The skin has a slight colour difference. It is the only visible sign of the treatment.”Female, 67

Another patient remarked,

“The CAEP treatment is an easy solution. It does not hurt. It is not complicated. There are no side effects and no medication. Only the treatment. Nothing else.”Female, 71

### 3.9. Health-Related Quality of Life

In our study, social isolation due to COVID-19 pandemic restrictions was noted. However, despite these restrictions, participants’ health-related quality of life (QoL) increased following CaEP treatment. CaEP generally improved patient QoL two months after treatment (see Figure 3). An increase in health-related QoL correlated with successful treatment and symptom relief. Patients who received home nursing for tumour wound care prior to CaEP treatment were able to continue doing so. Pain relief led to better sleep and improved energy. All seven patients reported being able to resume activities that they had been unable to do due to pain, discomfort, cosmetic distress or bleeding.

“Now I am going shopping again—no worries about anything. It does not itch anymore. It no longer bleeds as I wash my hair and I can shower without worrying.”Female, 68

One reported:

“… I can use a bra and breast prosthesis without having pain. Now that I can wear my breast prosthesis, I have no reason to opt-out of being amongst other people. I can sleep coherently again without waking up in pain.Female, 63

Surviving patients who had tumours that responded to treatment after two months also had a positive effect after one year. Of the three interviewed patients, two were retreated during this time. Data after one year showed no apparent difference in the three interviewed patients’ health-related QoL compared to two months after CaEP treatment.

## 4. Discussion

This is the first qualitative investigation of quality of life and patient experience in calcium electroporation patients.

Before beginning therapy with CaEP, every patient had positive expectations that the treatment would work and that potential side effects would not affect their quality of life. Patients often anticipate cancer therapy to be especially uncomfortable or worse than it is [26]. It has been demonstrated that the public allocates varying statuses to different types of cancer, with some tumours being regarded more highly than others [27].

Even though patients hoped the treatment would be helpful, most worried about the sensation and any harmful consequences of the electrical shocks. CaEP is new and unfamiliar to most; therefore, participants requested peer perspective, particularly between the preliminary evaluation and treatment. This study therefore suggests that personal accounts are important for patients prior to treatment, as seen in other settings [28].

This study investigated a small cohort from a single cancer centre. The participants differed in gender, cancer type and clinical cancer stage. All patients lived with spouses and spent time with family and friends regularly. Interdisciplinary discussion of empirical data helped qualify relevant everyday phenomena that affected individuals’ QoL.

Wenger et al. describe QoL as “perceptions of a person’s functioning and well-being” [29] and feeling well about oneself as well as others [30]. The therapeutic approach should consider phenomena such as wound symptoms and aesthetics as important in everyday life. From the patient’s perspective, healthcare professionals can learn what is essential from assessment through follow-up and incorporate it into shared decision-making. Shared decision-making is particularly important when planning calcium electroporation treatment. There may be weeks of healing needed after treatment, which both patients and practitioners must take into account. In addition, if the target area(s) cannot be treated with local anesthesia in one session (e.g., if the tumours are bulky, numerous or recurrent), multiple CaEP sessions may be necessary. It is also important to note that CaEP treatment has the benefit of being applicable even if patients are undergoing concurrent oncological treatments. Our findings can be used by medical professionals to educate individuals with cutaneous metastases about calcium electroporation. The patient quotes and experiences shared in this paper may also be helpful to future patients. As shown in Figure 3, almost all (n = 7) participants reported improvement in direct and indirect symptoms related to cutaneous metastases two months after CaEP. There were two reports of temporary effects on fatigue and vigour, one report of unchanged oozing and one report of unchanged fear of death. Successful treatment had a positive effect on pain, symptomatic wounds, sleep, vigour and social inclination. These findings support the idea that CaEP should be offered to eligible patients, as symptoms that affect quality of life may improve and, importantly, are unlikely to worsen.

## 5. Conclusions

This is the first study to use qualitative interviews and analysis to investigate the quality of life and experiences of patients undergoing CaEP treatment. When preparing for treatment, patients sought out the opinions and experiences of other peers. Patients had a positive outlook on calcium treatment with concerns about electrical pulses that exceeded their actual experience. Overall, CaEP treatment of cutaneous metastases improved patients’ quality of life by reducing pain and cosmetic distress, improving sleep and increasing vigour and social inclination.

## Figures and Tables

**Figure 1 cancers-15-00599-f001:**
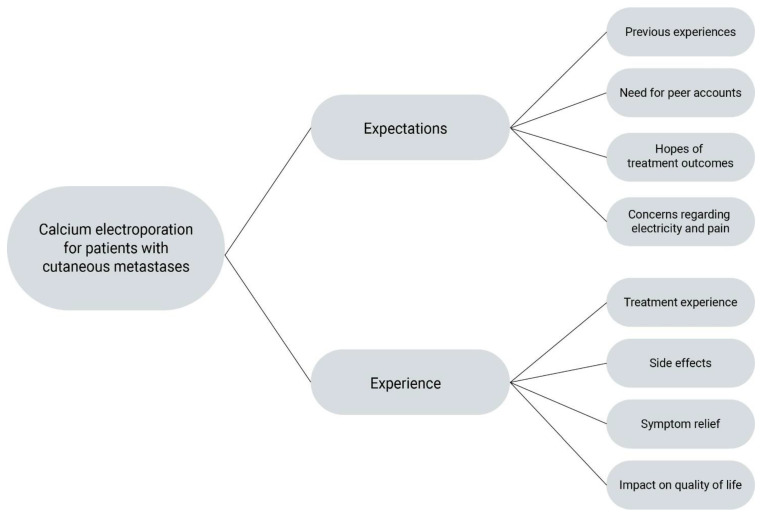
Themes and subthemes.

**Figure 2 cancers-15-00599-f002:**
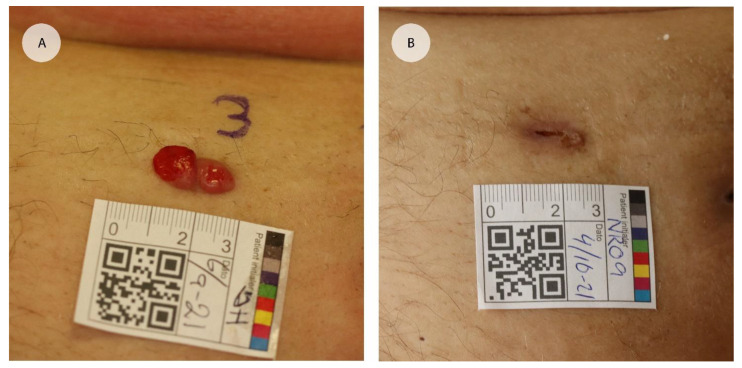
Example of affect in an endometrial cancer metastasis treated with calcium electroporation. (**A**) Digital clinical photograph at baseline before one calcium electroporation treatment in local anaesthesia. (**B**) One month after calcium electroporation treatment.

**Figure 3 cancers-15-00599-f003:**
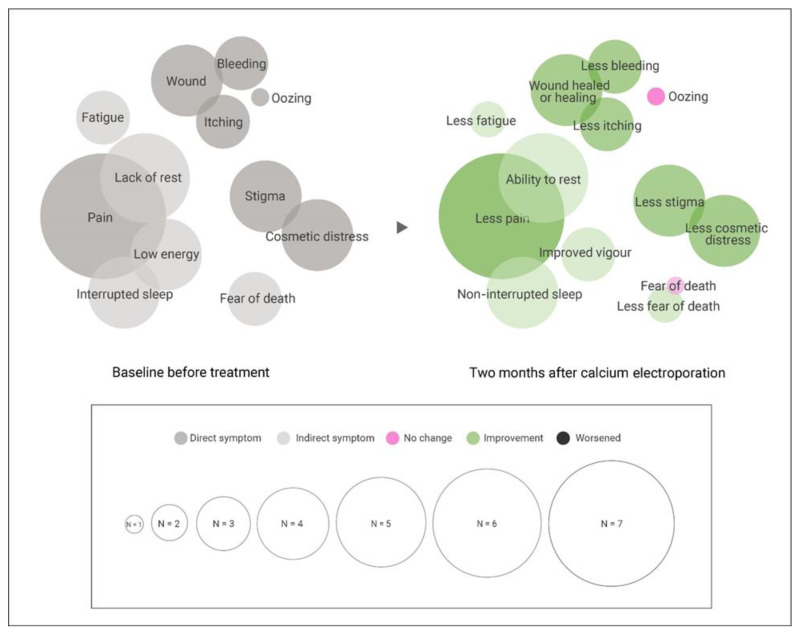
Patient-reported symptoms related to cutaneous metastases that affect quality of life before and two months after calcium electroporation. Bubble size is diameter relative to number of reporting patients from qualitative interview data, with the smallest bubble representing one report (n = 1) and largest seven reports (all, n = 7). Intense colour depicts direct symptoms from cutaneous metastasis, and low intensity depicts indirect symptoms. Green indicates improved symptoms. Note one patient reported no improvement in fear of death, and one patient reported unimproved oozing after two months (pink bubbles). The reporting patients of all other data points are the same before and after. One patient experienced transient improvement of fatigue, and one experienced transient improved vigour; These reports were not counted as improved or unchanged after two months. Apart from the two reports of unchanged symptoms and two reports of transient effects, there was unanimous improvement of direct and indirect symptoms related to cutaneous metastases.

**Table 1 cancers-15-00599-t001:** Demographics of patients treated with calcium electroporation interviewed at baseline and two months.

Pt. No.	Sex	Age ^1^	Household	Occupation	Daily Activities	Treatment Target	Primary Disease	Tumour Symptoms	Other Disease Symptoms
1	F	71	Spouse	Farmer (retired)	Walking the dog, spending time with family, reading, exercise	Metastases on chest	Breast cancer	Anxiety, pain	Back pain, anxiety (fear of death)
2	F	67	Spouse	Unemployed	Short activities, spending time with family	Metastasis on back	Lung cancer	Pain	COPD*, anxiety (fear of death and pain)
3	M	66	Spouse	Plumber	Full time work, outdoor activities, spending time with family	Metastases in flank	Lung cancer	Pain, interrupted sleep	Weakness, breathlessness
4	F	63	Spouse	Optician	Housework, gardening, part-time job, spending time with family	Metastases on the chest	Breast cancer	Pain, interrupted sleep, fatigue	Joint pain, anxiety, cosmetic distress
5	F	68	Spouse	Restaurant manager	At-home exercise, spending time with family	Metastases in flank	Lung cancer	Cosmetic distress, anxiety	Disturbed smell and taste, nausea and vomiting
6	F	68	Spouse	Office assistant	Spending time with family	Metastases on the head, neck and trunk	Breast cancer	Pain, bleeding, itching, cosmetic distress	Hip pain, walking disability, anxiety (fear of death)
7	F	62	Spouse	School teacher (retired)	Spending time with family	Metastases in the pubic and left inguinal area	Endometrial cancer	Cosmetic distress, anxiety, bleeding, oozing	Depression

^1^ Age in years. * COPD: Chronic obstructive pulmonary disease

## Data Availability

In accordance with the Danish Data Protection Agency, all recordings and transcripts of interviews will be deleted August 2035. Data-sharing requires permission from relevant authorities.

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
