# Peer review of "Qualitative Investigation of Experience and Quality of Life in Patients Treated with Calcium Electroporation for Cutaneous Metastases"

_cancers, 2023, doi:10.3390/cancers15030599_

Round 1

Reviewer 1 Report

Dear authors!

Your paper is interesting, understandable and clearly written. There are only a few minor issues that should be corrected (in my opinion):

1. Please substitute the sentence Approximately 5 to 20% of breast cancer patients develop cutaneous metastases in simple summary (line 17, page 1) with more general statement.  In your study you've dealt also with other types of metastatic cancers, not only breast metastases. Please give the general description of incidence of skin metastases and tumor types that are most often involved into dissemination to the skin.

2. Literature is not so scarce about cutaneous metastases (There is a scarcity of literature on cutaneous metastases, particularly on patient experience of living with cutaneous metastases (lines 21,22 page 1) . I would suggest to erase the first part of the sentence and to leave the part about scarcity of the literature on patient experience of living with cutaneous metastases.

3. Line 52 page 2. I would suggest to start the statement For example, approximately 5-10%... and to add the frequency of lung cancer cutaneous metastases since there is a same number of breast and lung cancer metastases treated.

4. Line 56 page 2. same comment as 2.

Author Response

Dear reviewer!

Thank you for your beneficial advice on the manuscript script. Please see the changes below.

It is our hope that the changes are in satisfaction to your recommendations.

We appreciate your feedback and efforts in improving our manuscript.

Best wishes on behalf of the Author Group

Kitt Vestergaard [email protected]

Changes in the text:

  1. Please substitute the sentence Approximately 5 to 20% of breast cancer patients develop cutaneous metastases in simple summary (line 17, page 1) with more general statement.  In your study you've dealt also with other types of metastatic cancers, not only breast metastases. Please give the general description of incidence of skin metastases and tumor types that are most often involved into dissemination to the skin.

  • Author response: Thank you for this excellent suggestion. The text below demonstrates how the script incorporates the suggested change.
  • Action taken: In line 17 page 1, we have changed the text from:

"Approximately 5 to 20% of breast cancer patients develop cutaneous metastases. When cancer is present on the body's surface or on the face, symptoms and distress have a significant impact on the quality of life of patients. Surgical and medical treatment is difficult, and local treatments are important."

To (line 17-20):

"Some cancer patients develop cutaneous metastases at late stage of disease, and the tumours may be present for months or years. Some cancers are more prone to disseminate to the skin such as breast and lung cancer. When cancer manifests on the body's surface or on the face, symptoms and distress can have a significant impact on the quality of life of patients”

  1. Literature is not so scarce about cutaneous metastases (There is a scarcity of literature on cutaneous metastases, particularly on patient experience of living with cutaneous metastases (lines 21, 22 page 1) . I would suggest to erase the first part of the sentence and to leave the part about scarcity of the literature on patient experience of living with cutaneous metastases.

  • Author response: Thank you for this excellent suggestion. The text below demonstrates how the script incorporates the suggested change.
  • Action taken: In line 21-22 page 1, we have changed the text from:

“There is a scarcity of literature on cutaneous metastases, particularly on patient experience of living with cutaneous metastases.”

To (line 23):

“There is a scarcity of literature on patient experience of living with cutaneous metastases.”

  1. Line 52 page 2. I would suggest to start the statement For example, approximately 5-10%... and to add the frequency of lung cancer cutaneous metastases since there is a same number of breast and lung cancer metastases treated.

  • Author response: Thank you for this excellent suggestion. The text below demonstrates how the script incorporates the suggested change.
  • Action taken: : In line 52 page 2, we have changed the text from:

“Approximately 5 to 20% of breast cancer patients develop cutaneous metastases”

To (line 52-53 page 2):

“For example, cutaneous metastases occur in 5 to 20% of breast cancer patients and are the most common type of cutaneous metastasis in women”

  1. Line 56 page 2.: The same comment as 2.

  • Author response: Thank you for this excellent suggestion. The text below demonstrates how the script incorporates the suggested change.
  • Action taken: : In line 56 page 2, we have changed the text from:

“There is a scarcity of literature on cutaneous metastases, particularly on patient experience of living with cutaneous metastases.”

To (line 59-60 page 2):

“There is a scarcity of literature on cutaneous metastases, particularly on patient experience of living with cutaneous metastases”

In addition, we have changed the following text:

Line 404 page 10:

“KV was the primary author of the protocol and the manuscript. KV, MV, JG and CLL designed the study. KV, MV, JG, and CLL contributed to writing of the manuscript. All authors read and approved the final manuscript. JG and MV have selected and treated the patients. KV has prepared an interview guide and conducted all interview.”

To (line 430 page 435):

“KV was the primary author of the manuscript. MV was the primary author of the clinical protocol. Figures by KV and MV. KV, MV, JG and CLL designed the study. KV, MV, JG, and CLL contributed to writing of the manuscript. All authors read and approved the final manuscript. JG and MV have included and treated the patients. KV prepared the interview guide and conducted all interviews.”

We have add the following text:

Line 426- 428 page 11): “We appreciate the assistance of Student Stine Vestergaard Jacobsen, who transcribed the interviews.”

Reviewer 2 Report

This manuscript provides a collective knowledge about how patients feel after getting Calcium Electroporation treatment. This type of data is critical to understand the perspectives of patients and how the treatment impacts their quality of life. It is good to know patients experienced improved symptoms after a course of CaEP treatment. This is very promising and encouraging for physicians and researchers but also for developing a better treatment regimen in future. The paper is well written, arranged and clear. However, couple questions came up while reading through it.

In abstract, line 30, “…with cutaneous metastases patients” here patients can be removed as its already mentioned in line 29

Line 43, “Peer accounts are important in preparation for treatment” means the patients should get a chance glancing the accounts to gain confidence in treatment?

To make sure I understand it correctly, the patients were actively receiving treatment for primary tumor during this study was planned? Or was offered earlier?

Line 160, Only three patients interviewed after one year. Curious why the other 4 were not able to participate?

In general, Did the questions asked were able to distinguish the symptom experience they might have due to other treatments?

Overall, the collected qualitative data is very warming to read. Larger subject group would have been better, but the size included looks decent. Also, qualitative data collected significant time (like >1 year) after treatment might be more convincing. Hope this data will further assist in improvising the treatment according to the patient’s comfort.

Author Response

Dear reviewer!

Thank you for your beneficial advice on the manuscript script. Please see the changes below.

It is our hope that the changes are in satisfaction to your recommendations.

We appreciate your feedback and efforts in improving our manuscript.

Best wishes on behalf of the Author Group

Kitt Vestergaard [email protected]

Changes in the text:

  1. In abstract, line 30, “…with cutaneous metastases patients” here patients can be removed as its already mentioned in line 29.

  • Author response: Thank you for this excellent suggestion. The text below demonstrates how the script incorporates the suggested change.
  • Action taken: In line 30 page 1, we have changed the text from:

“Data on quality of life for patients with cutaneous metastases patients treated with calcium electroporation is limited”

To (line 30-31 page 1):

“Data on quality of life for patients with cutaneous metastases treated with calcium electroporation is limited”

  1. Line 43, “Peer accounts are important in preparation for treatment” means the patients should get a chance glancing the accounts to gain confidence in treatment?

  • Author response: Thank you for this excellent suggestion. The text below demonstrates how the script incorporates the suggested change.
  • Action taken: In line 43 page 1, we have changed the text from:

“Peer accounts are important in preparation for treatment.”

To (line 44-45 page 1):

“Peer accounts provide patients with a shortcut to confidence in treatment on top of doctors' recommendations.”

  1. To make sure I understand it correctly, the patients were actively receiving treatment for primary tumor during this study was planned? Or was offered earlier?

  • Author response: Thank you for this excellent suggestion. The text below demonstrates how the script incorporates the suggested change.
  • Action taken: In line 93-94 page 2, we have changed the text from:

“All patients received treatment, and calcium electroporation was not compared to other treatment modalities.”

To (line 96-103 page 3):

“The methods and materials used are thoroughly described in a protocol article, ClinicalTrials.gov Identifier: NCT04225767 [1]. Briefly, qualitative interviews were performed in a subset of patients treated in the CaEP-R study at Zealand University Hospital, Denmark. All patients received calcium electroporation treatment, and calcium electroporation was not compared to other treatment modalities. Patient could continue any concomitant systemic treatment. Interviews before- as well as two and twelve months after treatment were conducted from February 2020 to November 2021. The first interview took place after the preliminary examination, before CaEP, the day of treatment.”

  1. Line 160, Only three patients interviewed after one year. Curious why the other 4 were not able to participate?

  • Author response: Thank you for this excellent suggestion. The text below demonstrates how the script incorporates the suggested change.
  • Action taken: In line 159 page 4, we have changed the text from:

“Three patients were interviewed one year after treatment.

To (line 165-166 page 4):

“Three patients were interviewed one year after treatment, and four patients had passed away.”

  1. In general, did the questions asked were able to distinguish the symptom experience they might have due to other treatments?

  • Author response: Thank you for this excellent suggestion. The text below demonstrates how the script incorporates the suggested change.
  • Action taken: In line 123-124 page 2, we have changed the text from:

“The questions examined patients' emotional and physical well-being before and after CaEP treatment.”

To (line 128-130 page 3):

” The questions examined patients' emotional and physical well-being with emphasis on symptoms related to their cutaneous metastases, before and after CaEP treatment.”

In addition, we have changed the following text:

Line 404 page 10:

“KV was the primary author of the protocol and the manuscript. KV, MV, JG and CLL designed the study. KV, MV, JG, and CLL contributed to writing of the manuscript. All authors read and approved the final manuscript. JG and MV have selected and treated the patients. KV has prepared an interview guide and conducted all interview.”

To (line 430 page 435):

“KV was the primary author of the manuscript. MV was the primary author of the clinical protocol. Figures by KV and MV. KV, MV, JG and CLL designed the study. KV, MV, JG, and CLL contributed to writing of the manuscript. All authors read and approved the final manuscript. JG and MV have included and treated the patients. KV prepared the interview guide and conducted all interviews.”

We have add the following text:

Line 426- 428 page 11): “We appreciate the assistance of Student Stine Vestergaard Jacobsen, who transcribed the interviews.”